# Targeting Children and Their Mothers, Building Allies and Marginalising Opposition: An Analysis of Two Coca-Cola Public Relations Requests for Proposals

**DOI:** 10.3390/ijerph17010012

**Published:** 2019-12-18

**Authors:** Benjamin Wood, Gary Ruskin, Gary Sacks

**Affiliations:** 1Global Obesity Centre, Deakin University, Deakin University Waterfront Campus, 1 Gheringhap St, Geelong, VIC 3220, Australia; gary.sacks@deakin.edu.au; 2U.S. Right to Know, 4096 Piedmont Ave. #963, Oakland, CA 94611-5221, USA; gary.ruskin@gmail.com

**Keywords:** food industry, The Coca-Cola Company, public relations campaigns, childhood obesity, marketing to children, corporate political activity

## Abstract

The study provides direct evidence of the goals of food-industry-driven public relations (PR) campaigns. Two PR requests for proposals created for The Coca-Cola Company (Coke) were analysed. One campaign related to the 2016 Rio Olympic Games, the other related to the 2013–2014 Movement is Happiness campaign. Supplementary data were obtained from a search of business literature. The study found that Coke specifically targeted teenagers and their mothers as part of the two PR campaigns. Furthermore, Coke was explicit in its intentions to build allies, particularly with key media organisations, and to marginalise opposition. This study highlights how PR campaigns by large food companies can be used as vehicles for marketing to children, and for corporate political activity. Given the potential threats posed to populations’ health, the use of PR agencies by food companies warrants heightened scrutiny from the public-health community, and governments should explore policy action in this area.

## 1. Introduction

Large transnational companies in the food and beverage industry (hereinafter the food industry) employ a diverse range of strategies to protect and pursue their corporate interests [1]. Due to the extent to which these interests, including profits from sales of unhealthy foods and beverages, adversely impact public health, the food industry has commonly been labelled as a major driver of obesity and diet-related chronic diseases [2].

Companies in the food industry, like those in other major industry sectors, use public relations (PR) as a key means of deploying their corporate messages [1]. PR agencies are highly effective in designing and implementing the food industry’s use of integrated marketing communications (IMCs) [3], combining a variety of communication disciplines—public relations, general advertising, direct response, and sales promotion—to promote foods and beverages and their associated brands [4]. PR agencies are also well-placed to assist a company’s efforts to influence public policy and opinion in its favour [1]—conduct known as corporate political activity (CPA). While the use of PR campaigns is broadly accepted as a normal business practice, its use by food companies has been denounced by some public-health commentators as an attempt at “damage control” in light of the risks posed to industry profits by concerns about obesity and related diseases [1].

Although there is a large body of public-health literature surmising the aims of the food industry’s use of PR, there is limited direct evidence of the goals and purposes of food-industry-driven PR campaigns and the way in which food companies brief PR agencies. This study aimed to analyse, from a public-health perspective, The Coca-Cola Company (Coke)’s key PR targets and strategies from two recent PR campaigns.

## 2. Materials and Methods

On July 17, 2015, U.S. Right to Know (USRTK), a nonprofit investigative research group, sent a Colorado Open Records Act request to the University of Colorado for documents related to Coke, among other subjects. USRTK received 11,714 pages in response, all of which can be found online at the University of California, San Francisco’s (UCSF’s) Industry Documents Library. This included two PR requests for proposals (RFPs)—the only RFPs made available—created for separate Coke campaigns: the 2013–2014 Movement is Happiness campaign and the 2016 Rio de Janeiro Olympic Games campaign (the Rio Campaign). These can be found in the UCSF Food Industry Collection of the Food Industry Archive [5,6]. Data from the two RFPs were supplemented by data obtained from a search of business literature that discussed and analysed the campaigns. Both RFPs appeared to focus at the brand level rather than on specific products.

The two RFPs were qualitatively examined using content analysis guided by a framework approach. The stated targets and strategies deployed as part of the campaigns were identified and framed using an approach informed by the Mialon et al. CPA identification framework, and Nestle’s Big Soda’s PR Playbook [1,7]. Mialon et al.’s CPA identification framework categorises the corporate political activity of the food industry from a public-health perspective into the following groups: (i) information and messaging, (ii) financial incentives, (iii) constituency building, (iv) policy substitution, (v) legal strategies, and (vi) opposition fragmentation and destabilisation [7]. Nestle’s Big Soda’s PR Playbook thematically classifies the messages, methods and actions of the soda industry employed to achieve three corporate goals: avoid public criticism, oppose government regulation and protect sales [1]. The four parts of the playbook are to: (i) emphasise devotion to health and wellness; (ii) sell to everyone, everywhere; (iii) build allies and create conflicts of interest among potential critics; and (iv) ‘play hardball’ and protect corporate interests [1].

## 3. Results

The two RFPs provide direct evidence of Coke’s: (i) intended targets; and (ii) intended use of two key CPA strategies: constituency building and marginalising opposition.

### 3.1. Key Targets

The Rio Campaign directly set out to target teenagers, mothers, “influencers” (particularly on social media) and celebrities: “The ideal [campaign] would target: teens, moms, influencer/celebrities, global media (teen, industry)”.

Campaign US—the US edition of the global business magazine Campaign—lauded the effectiveness of Coke’s Rio Campaign in reaching teenagers (defined by Coke as persons aged 13 to 20). They supported this statement with reference to the following marketing indicators [8]:The campaign reached 21 million teenagers (90% on mobile phone platforms) with an 88% accuracy rate for targeting teenagers;The phase of the campaign during the games achieved a seven-point brand lift (a measurement of the increase in brand interaction) for teenagers.

To help Coke engage with teenagers, the Rio Campaign enlisted a number of key global influencers and celebrities. These included social media personality Jake Boys, Radio Disney star Alli Simpson and several well-known members of the United States (US) Olympic team [9].

The Movement is Happiness Campaign also focused on teenagers, with an emphasis on increasing the perceived healthiness of the Coke brand: “[Coke] is interested in engaging an agency […] to achieve the following goals: […] Increase Coke brand health scores with teens.”

### 3.2. Constituency Building and Marginalising Opposition

Coke was explicit in its intent to use the Movement is Happiness campaign to build key allies and marginalise opposition: “[Coke] recognizes the importance of leading in this [health and well-being] space to marginalize detractors and build support broadly in a host of categories, including consumers, women, government and political officials and personalities, and media.”

Specifically, a common directive from Coke was to establish relationships with the media:
Continue to build strong relationships with key marketing journalists via face-to-face meetings to ensure [Coke] is front-of-mind as an industry “go-to” and to help negate negative media coverage.
Identify 1–2 moments in which [Coke] can look to engage our key media influencers across marketing trades to build existing and create new relationships with media.
[Key performance indicators] should capture relationship briefings with key media […] and successful liaison with other PR agencies to source information and coordinate news stories.

## 4. Discussion

This study provided direct evidence that Coke intended for two of its recent PR campaigns to target teenagers and their mothers. In addition, the findings supplement and support previous analyses that have demonstrated Coke’s use of constituency building, including the intent to establish relationships with, and influence, key media, as well as opposition marginalisation strategies [1,10,11,12].

The large number of children targeted and reached by Coke as part of their PR campaigns is a serious concern from a public-health perspective. While Coke explicitly wrote that it aimed to target teenagers as part of the analysed campaigns, the enlistment of celebrities with fan bases inclusive of children under 12 almost certainly means that the campaigns also resulted in exposing younger children to its messaging [13,14]. Coke pledges not to directly target marketing at children under 12 by not placing advertising in media where more than 35% of the audience is children under 12 [15]. Coke also states that it is working with several industry platforms—the International Food & Beverage Alliance, the World Federation of Advertisers, the European Union Pledge Initiative and the Children’s Food and Beverage Advertising Initiative—to “reduce the exposure of children to advertising of all products high in fat, salt or sugar” through “meaningful actions” [15]. Our analysis of these two PR campaigns clearly demonstrates the limitations of Coke’s pledges and its “Responsible Marketing” policy in general. To highlight this point, if the overall number of children and teenagers reached by an advertising campaign is 21 million (the number reached by the Rio campaign), then according to Coke’s policy it would be acceptable for more than 7 million of these to be below the age of 12. 

Coke’s marketing strategy to directly target mothers—a tactic described as one way to side-step the issue of advertising directly to young children [4]—also demands further scrutiny from a public-health perspective.

The key strength of this paper is that it analyses documents created by a major food company, providing direct evidence of its PR campaign targets and strategies. Thus, this paper adds to the limited existing evidence in the public-health literature derived from internal food-industry documents [11,12,16,17]. Important limitations of this paper are that the analysis only involved Coke’s RFPs for two PR campaigns, and that it only superficially examined how the campaigns were executed. The reasoning behind the latter limitation is that the paper was not intended to measure the effectiveness of the Coke PR campaigns, but instead focus on the intentions of the campaigns discussed ‘behind closed doors.’

## 5. Conclusions

Coke’s intent and ability to use PR campaigns to market to children should cause serious public-health concern, given that the exposure of children to the marketing of unhealthy foods is likely to be an important contributor to increased childhood obesity rates [18]. Adding to this concern is that Coke’s PR campaigns can influence millions of children through a marketing approach that: (i) uses multiple communication channels (notably social media) to reach children; and (ii) is becoming more subtle, with marketing techniques integrated into traditionally non-promotional opportunities [4].

Ultimately, government policy is needed to effectively restrict the exposure of children to the marketing of unhealthy foods. Given the role of PR agencies in designing and implementing integrated marketing communications (IMC), their use by food companies should fall within the reach of such policy or regulatory action.

More broadly, global action is needed to eliminate the advertising and marketing of harmful products, including unhealthy food products, at sporting events [19,20]. This would require global sporting bodies such as the International Olympic Committee and FIFA to play their parts in addressing the issue at hand [19].

The public-health community should also increase its scrutiny of the use of PR agencies by food companies beyond the advertising and marketing dimensions. PR agencies can be a potent vehicle for corporate political activity. Their use therefore warrants careful monitoring to better understand the power dynamics at play between corporate actors and public-health interests.

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
