# Peer review of "Targeting Children and Their Mothers, Building Allies and Marginalising Opposition: An Analysis of Two Coca-Cola Public Relations Requests for Proposals"

_ijerph, 2019, doi:10.3390/ijerph17010012_

Round 1
Reviewer 1 Report
This is an interesting and nicely written paper with potential social and policy implications. It is a very brief (I guess there is a reason it is so short? Is it a Research note?) look at the RFPs of Coke according to two campaigns - Rio and Happiness.The former is addressed in more detail because it has some evaluative material via Campaign US. Can the authors add similar information for the latter RFP? If not, I wonder if this case is worth keeping in, because it is dealt with much more briefly than the Rio example, and it is also quite dated now. Can the authors also say why these two specific cases were selected and from how many?
What products were involved in the RFPs or were they at a generic brand level?
For readers unfamilar with Mialon et al and Nestle's Bog Soda Playbook, it would be helpful to say why these were used and how.
What is the evidence for the claim on p3 line 100 that the celebrities have fan bases with children under 12?
Could the authors add an implications section? It is left quite vague as to what impact this research is expected to have ('explore policy action in this area' p3 line 122).
I hope these comments will make the brief article more impactful.
Good luck with your research.
Reviewer 2 Report
This is an excellent paper that reveals the 'hidden' strategies and tactics of a major multinational company. While the results are no surprise in some ways given the history of the marketing of unhealthy food and beverage to children and mothers, in others they are groundbreaking. The authors are to be congratulated for a well-written and innovative paper.
I endorse acceptance of this paper.
Reviewer 3 Report
This paper investigates how Coca Cola’s recent two PR campaigns target teenagers and their mothers. Although it is important to explore how food companies employ their PR campaigns to target children, this paper does not provide insightful implications for the academia and policy makers. Moreover, the method adopted in this paper did not produce convincing findings. The description of the method section is very general and vague. The authors should explicitly explain how they analyze the text of the two campaign proposals.
Round 2
Reviewer 3 Report
The authors have improved the manuscript by providing implications for policymakers. Furthermore, the authors have elaborated more in the method section. After the revision, the manuscript reads quite well.
The authors may consider citing some papers from IJERPH or relevant journals to link this paper to the field of public health.
Author Response
Thanks for your feedback. Additional refs were added in the text as per your suggestion. These have been highlighted in the manuscript (red text). We trust this paper is now acceptable for publication.
Round 3
Reviewer 3 Report
Good luck with future research